

# Phytochemical screening and *in vitro* antibacterial activity of *Echinops kebericho* Mesfin tuber extracts: experimental studies

Jiregna Gari Negasa[1], Ibsa Teshome[2], Edilu Jorga Sarba[3] and Bekiyad Shasho Daro[4]

[1] Department of Veterinary Laboratory Technology, School of Veterinary Medicine, Ambo University, Ambo, Oromia, Ethiopia
[2] School of Veterinary Medicine, Ambo University, Ambo, Oromia, Ethiopia
[3] Department of Veterinary Science, Ambo University, Ambo, Oromia, Ethiopia
[4] Department of Veterinary Epidemiology, College of Veterinary Medicine, Haramaya University, Dire dawa, Oromia, Ethiopia

## ABSTRACT

**Background**. The application of plant extracts and their phytochemicals as potential treatments for bacterial illnesses has increased significantly in the last few decades. In Ethiopia, *Echinops kebericho* Mesfin is widely used to treat a range of illnesses in humans and animals. This study aimed to evaluate the antibacterial activity and phytochemical screening of *Echinops kebericho* Mesfin.

**Methods**. We carried out an *in vitro* experimental study after collecting the plants from their natural habitats. Then macerated in absolute methanol and petroleum ether solvents and concentrated the extracts using a rotary evaporator. In the experiment, we used Standard cultures of *E. coli*, *K. pneumoniae*, *S. aureus*, and *P. aeruginosa*. The agar-well diffusion method evaluated the antibacterial activity of the plants. The agar dilution method determined the minimum inhibitory concentration of the plant extract.

**Results**. The percentage yield of the plant extracts ranged from 6.25% to 7.85%. The methanol extract of *Echinops kebericho* Mesfin had the highest inhibitory effect on *S. aureus* (ATCC 25923) (16.67 ± 0.58 mm), followed by *E. coli* (ATCC 25922) (11.0 ± 1.73 mm). Phytochemical screening of leaves from the methanol and petroleum ether extracts of the plant revealed the presence of phytochemicals such as alkaloids, flavonoids, tannins, and cardiac glycosides. The present study revealed that the extracts of these plants have antibacterial activity. However, researchers should conduct further studies on the safety margin and quantitative bioactive isolation of selected medicinal plants.

# INTRODUCTION

Throughout history, humans have utilized medicinal plants for medical purposes and have now derived many modern medications from them (*Jamshidi-Kia, Lorigooini & Amini-Khoei, 2017*). Medicinal plants are very vital in their uses for the treatment of

Corresponding author
Jiregna Gari Negasa,
jiregnagari@gmail.com

human and livestock ailments, besides providing ecological, economic, and cultural services. In developing countries, plants are the main source of treatment for people (*Van Andel & Carvalheiro, 2013*). Humans have used plants to treat common infectious diseases and still regularly use certain traditional medicines to treat a variety of illnesses (*Gurib-Fakim, 2006*). Numerous countries have a significant desire to find new ways to address antimicrobial resistance by actively utilizing their ethnopharmacological tradition to find new medications (*Li et al., 2022*).

Over the previous few decades, there has been a notable increase in the emergence of infectious bacterial pathogens that are resistant to antibiotics. Multidrug-resistant bacteria have made many of the currently prescribed antibiotics more difficult to obtain and less affordable globally (*Theuretzbacher, 2013*). Due to antimicrobial resistance (AMR), infectious diseases are emerging as major worldwide health concerns, resulting in millions of infections and thousands of deaths each year (*Ferri et al., 2017*).

To address the serious and challenging issue of antimicrobial resistance, which affects global health, a multidisciplinary approach involving partners from all health sectors— including public health authorities and the scientific community is necessary (*Salam et al., 2023*). Common infections have become incurable due to the rapid growth of antimicrobial resistance (AMR), which is outpacing the development of new drugs or alternative methods to combat them. There is an urgent need to find new antimicrobial agents and alternative ways to address this issue in light of this global resurgence (*Moo et al., 2020*).

The use of plant-based pharmaceuticals in place of contemporary treatments is expanding throughout the world's medical marketplaces (*Cordell, 2014*). Plants, which have long been used to make medicines, can be a source for the creation of novel antimicrobial compounds. These compounds may be the subject of further chemical, pharmacological, and clinical research (*Anand et al., 2019*). These medicines have played a significant role in improving human health. In search of potent antimicrobial agents for the treatment of illnesses caused by bacteria and other multidrug-resistant pathogens, medicinal plants are drawing the attention of scientists and researchers in the field of herbal medicine (*Chowdhury, Misra & Mandal, 2021*).

Plant materials may undergo extraction utilizing various techniques. Nonconventional methods have a lower environmental impact since they utilize fewer synthetic and organic chemicals, require less time to operate, and provide an extract with a higher quality and yield (*Jha & Sit, 2022*). Non-conventional extraction techniques, such as Microwave-Assisted extraction, Ultrasound-Assisted extraction, pressurized liquid extraction and supercritical fluid extraction have been researched to improve the overall yield and selectivity of bioactive components from plant sources (*Yolci Omeroglu et al., 2019*).

Many phytochemicals found in plants, including polyphenols, flavonoids, terpenoids, alkaloids, and tannins, have shown to have antimicrobial qualities and are used to treat viral infections, cancer, heart disease, and circulatory disorders (*Barbieri et al., 2017*). For thousands of years, Ethiopians have utilized the herb *Echinops kebericho* for medicinal purposes (*Demie, 2019*). In Ethiopia, *Echinops kebericho* Mesfin is an endemic medicinal plant that has used to treat a variety of infectious diseases. The plant was administered by mouth, inhalation, or topically applied to the affected area (*Ameya, Gure & Dessalegn,*

*2016*). *E. kebericho* Mesfin is widely used to treat a variety of illnesses in humans and animals, including gonorrhea, respiratory discomfort (*Dubale et al., 2023*), febrile illness, lung tuberculosis, trypanosomiasis, typhoid, typhus, colic, cough, scabies, malaria, headaches, fumigation during childbirth, and wound infections (*Deyno et al., 2020*).

In ethnoveterinary medicine, *E. kebericho's* tubers used to treat skin infections, respiratory symptoms, and blackleg disease in animals (*Tafesse et al., 2023*). Numerous pharmacological effects, including anthelminthic, antibacterial, antifungal, antidiarrheal, antispasmolytic, and antimalarial effects, have been demonstrated by the extracts of *E. kebericho* Mesfin. Methanolic tuber extracts from *E. kebericho* Mesfin demonstrated relatively mild antibacterial activity (*Ameya, Gure & Dessalegn, 2016*; *Biruksew et al., 2018*). Notable actions against the fungi *Aspergillus flavus* and *Candida albicans* also been documented (*Beressa, Deyno & Alele, 2020*). This plant shows antihelminthic properties against Trypanosoma, Leishmania, and malaria parasites (*Ameya, Gure & Dessalegn, 2016*; *Tariku et al., 2011*).

The contents of *Echinops kebericho* include steroids, alkaloids, triterpenoids, resins, and flavonoids (*Deyno et al., 2020*). The tubers of the plant were used to isolate and identify dehydrocostus lactone (*Tariku et al., 2011*; *Deyno et al., 2021*). Significant antimicrobial resistance-modulatory effects were observed for the isolation from the dichloromethane fraction. Dehydrocostus lactone, isolated from *E. kebericho*, reported having antimycobacterial activity against *Mycobacterium smegmatis* (*Deyno et al., 2020*). The antibacterial activity and phytochemical contents of *Echinops kebericho* Mesfin have not been well studied in the study area. However, they are commonly used by traditional healers to treat wound infections, toothache, tonsillitis, gonorrhea, respiratory manifestations, febrile illness, lung tuberculosis, trypanosomiasis, typhoid, typhus, common cold, cancer, colic, cough, scabies, malaria, headache, fumigation during childbirth, and as mosquito repellent (*Deyno et al., 2020*; *Tafesse et al., 2023*). Therefore, screening the antimicrobial activities of these plants is important.

# MATERIALS & METHODS

## Description of the plant collection areas

We conducted the study at Ambo University after collecting plants from two districts of the West Shewa zone of the Oromia regional state in Ethiopia, namely, the Ambo and Toke Kutaye districts. Administratively, the Ambo district belongs to the West Shewa Zone, Oromia Regional State of Ethiopia. The Ambo district is located at a latitude of 8°59′N and 8°98.3′N and longitudes of 37°51′E and 37°85′E with an elevation range of 1,900–2,275 m above sea level. The Toke Kutaye district is located in the west Shewa zone of the Oromia regional state, 126 kilometers west of Addis Ababa and 12 km from Ambo. Geographically, it is situated between latitudes 8°47′N and 9°21′N and longitudes 37°32′E and 37°03′E.

We designed the *in vitro* experimental study to be carried out from November 2022 to September 2023. All the experimental tests were performed in triplicate along with the positive and negative controls to determine the antibacterial activity of the plant.

## Plant material collection and identification

Mr. Fikadu Lebeta, a botanist in the Plant Science Department at Ambo University's Guder Mamo Mezemir Campus, collected tubers of *E. kebericho Mesfin* from the Toke Kutaye and Ambo districts and authenticated them. He collected and transported the plant materials using standard storage protocols and wrapped them in plastic sheets. Then, he deposited voucher numbers (AU2202) on the specimens of *E. kebericho* Mesfin in the Ambo University plant science laboratory. We washed tubers of *E. kebericho* Mesfin plants with clean water to remove dirt and rinsed them with distilled water. We cut the rinsed plant materials into pieces and dried them in the shade at room temperature. We ground the dried plant samples using a mortar and pestle. We stored powders of the plant material in clean and dry plastic bags.

## Plant extraction

We extracted the powder part of the study plant material utilizing absolute methanol (99.8%) and petroleum ether (99.8%) through the maceration method. They prepared the crude plant extracts separately using methanol (99.8%) and petroleum ether (99.8%). A sensitive digital weighing balance weighed out two hundred grams of air-dried powdered plant material. Then, it placed the material in Erlenmeyer flasks and filled them with 800 mL of extraction solvent (absolute methanol). The plant material was macerated for 72 h over an HZ rotary shaker at 121 rpm at room temperature, following the method described by *Crăciun & Gutt (2023)* for the solid–liquid ratio of plant material and solvent. The extract was separated from the marc using gauze, and the resulting suspension was filtered with Whatman™ filter paper number 1. The resulting filtrate was concentrated under reduced pressure in a rotary evaporator at 40 °C and 121 rpm, followed by heating in an oven at 40 °C (*Kifle, Anteneh & Atnafie, 2020*; *Marami et al., 2021*).

We soaked approximately 200 g of sieved plant powder in petroleum ether (99.8%) in separate 800 mL Erlenmeyer flasks on an HZ rotary shaker at 121 rpm at room temperature for 72 h (*Abubakar & Haque, 2020*; *Ali & Awoke, 2021*). The extracts were filtered using gauze followed by Whatman™ filter paper number 1. Concentrated the filtrate under reduced pressure in a rotary evaporator at 40 °C and 121 rpm, and then heated it in a hot air oven at 40 °C (*Yasien et al., 2022*). Transferred the dried crude extract of each solvent to a vial, measured the concentration of each plant extract, and stored the sample in a refrigerator until further processing. The percentage yields of all the plant extracts calculated using the following formula (*Abbas et al., 2021*; *Gizaw et al., 2022*).

$$\text{Yield}(\%) := \frac{\text{Weights of extract}\,(\text{g})}{\text{Weight of plant material}\,(\text{g})} * 100.$$

## Preliminary phytochemical screening of the extract

We conducted a standard preliminary phytochemical analysis to determine the presence or absence of secondary metabolites in the methanol and petroleum ether crude extracts of each plant using standard phytochemical screening methods. We identified the presence of bioactive compounds such as flavonoids, saponins, alkaloids, tannins, phenols, cardiac

glycosides, and phytosterols in crude tuber extracts of *E. kebericho* through standard testing methods.

## Test for alkaloids

Wagner's reagent used to test the alkaloid content. Briefly, 15 mg of crude plant extract was mixed from each of the two solvents in a water bath for 5 min before filtering. A few drops of Wagner's reagent were added to the plant extract filtrates, and the presence of brown or reddish precipitates indicated the presence of alkaloids in the extract (*Joshi, Bhobe & Sattarkar, 2013*; *Iqbal, Salim & Lim, 2015*).

## Test for flavonoids

A portion of the powdered crude plant (0.25 g) sample was heated for 3 min in the presence of ten milliliters of ethyl acetate for every plant extract. After the mixtures were filtered, 1 ml of diluted ammonia solution was shaken with 4 ml of the filtrate. The solution was yellow, which is indicative of the presence of flavonoids (*Edeoga, Okwu & Mbaebie, 2005*; *Lalitha & Gayathiri, 2013*).

## Test for saponins

We boiled approximately 2 g of the crude sample extract in 20 ml of distilled water in a water bath and filtered it. They then mixed 10 milliliters of the filtrate with 5 ml of distilled water, shook it vigorously for 2–3 min, and checked for stable persistent froth, which indicated saponins in the sample extract (*Edeoga, Okwu & Mbaebie, 2005*; *Dhayalan et al., 2018*)

## Tannin test

We used a bromine water test to determine the tannins in the plant extracts. They dissolved 5 mg of the crude extracts from the plants in 5 ml of 95% ethanol and filtered the solution. They then added three or four drops of bromine water to the filtered solution. They detected condensed tannins in the solution by observing buff-colored precipitates; the absence of color showed hydrolyzable tannins (*Ali et al., 2018*; *Salehdeen, Abdulrazaq & Osinlu, 2019*).

## Test for phenols

Phenol was detected in the plant extracts using the ferric chloride test. Briefly, 0.25 gm of the crude extract was treated with 4 drops of $FeCl_3$ to afford a blue −black color, which indicates the presence of phenols in the plant extracts (*Gizaw et al., 2022*; *Hasan et al., 2017*).

## Test for phytosterols

We applied chloroform to 0.25 grams of plant extract and filtered the resulting solution to identify the presence of phytosterols through the Szarkowski test. The filtrates were mixed with concentrated $H_2SO_4$, vigorously shaken, and then allowed to stand for a brief period of time. The ability of the extracts to produce a golden yellow hue suggested the presence of phytosterols (*Abubakar & Haque, 2020*; *Hasan et al., 2017*).

## Tests for cardiac glycosides

We tested the extract for cardiac glycosides using the Keller-Killani test. We dissolved 0.2 grams of the extract by mixing two milliliters of glacial acetic acid with one drop of a 5% ferric chloride solution. We filled the test tube containing the solution with a few drops of concentrated $H_2SO_4$. A cardiac glycoside is present in a solution that gradually becomes bluish-green in color (*Oloya et al., 2022*).

## Bacterial strains

The standard reference bacterial species used were the American Type Culture Collection (ATCC) of *Escherichia coli* (ATCC 25922), *Klebsiella pneumoniae* (ATCC 43816), *Staphylococcus aureus* (ATCC 25923) and *Pseudomonas auroginosa auroginosa* (ATCC 27853), which were collected from the Bacteriology Laboratory of the Ethiopian Public Health Institute (EPHI). We cultured and maintained these bacterial strains in appropriate culture media before performing antimicrobial tests. We performed all laboratory work according to CLSI guidelines (*CLSI, 2022*).

## Determination of the antibacterial activity of the extracts
### Preparation of working stock solution

We prepared a stock solution of the plant extract by dissolving 1 g of extract in 5 ml of 5% DMSO to get 200 mg/ml of each plant extract. We selected DMSO (5%) because it can dissolve these plants. Then, we serially diluted the solution to get 100 mg/ml, 50 mg/ml, and 25 mg/ml solutions.

## Inoculum preparation

Differential and selective media were used to streak standard test organisms. Bacterial strains were verified and confirmed after biochemical analysis, Gram staining and growth on selective media. We used mannitol salt agar for *S. aureus*, eosin-methylene blue (EMB) agar for *E. coli*, cetrimide agar for *P. aeruginosa*, and MacConkey agar for *K. pneumoniae*. We prepared the inoculum according to the Clinical and Laboratory Standards Institute (CLSI) guidelines and standardized it to a 0.5 McFarland turbidity standard equivalent. The bacterial strains were cultured on selective media or differential media at 37 °C for 24 h. We transferred sterile nutrient agar plates to the colonies for subculture. We transferred sterile nutrient agar plates to the colonies for subculture. Four to five bacterial colonies removed from the subcultured agar and inoculated into a sterile physiological saline solution (*Cavaleri et al., 2005*). We emulsified the colony suspensions in sterile saline and diluted them until the turbidity matched that of a 0.5 McFarland turbidity standard, with a concentration of ($1.5 \times 108$ CFU/ml) (CLSI, 2012 A Wickerham Card was used to visualize turbidity regarding the 0.5 McFarland turbidity standard after the suspension was vortex mixed (*Fadahunsi & Babalola, 2021*).

## Agar well diffusion

The agar well diffusion method used to evaluate the antimicrobial activity of the plant extracts. Agar well diffusion method was used to evaluating the extracts antibacterial activity. The media cooled immediately after autoclaving, at temperatures between 45 to

50 °C. The freshly prepared and cooled media were poured into 90 mm diameter Petri dishes and set on a level and horizontal surface to achieve a uniform depth of nearly four mm. The agar media cooled and solidified at room temperature (*Cavaleri et al., 2005*).

McFarland standard (0.5) used to adjust vortexed bacterial suspension (equivalent to $1.5 \times 10^8$ CFU/ml) by contrast against Wickerham Card (black and white) stripped paper to visualize turbidity to the standard level. The bacterial suspension was prepared in a sterile physiological saline solution tube within 15 min. The aliquot spread evenly onto MHA with a sterile cotton swab (*Wiegand, Hilpert & Hancock, 2008*).

The swabbed Mueller Hinton agar was left for 15 min to allow bacteria to attach to the media. A sterilized cork borer 6 mm in diameter was used to create a well on swabbed MHA medium to form 6 mm diameter wells. Five equidistant wells 6 mm in diameter and 4 mm in depth were made on the agar plate. With a sterile needle, we discarded the circle created by the cork borer to make a well. The created wells filled with 50 μL of plant extracts at different concentrations (200 mg/ml, 100 mg/ml, 50 mg/ml, and 25 mg/ml) or with 5% DMSO as a negative control (*Jamal et al., 2011*; *Manandhar, Luitel & Dahal, 2019*). After all the agar wells were filled and a positive control was placed, the plates were kept at room temperature for approximately 40 min to allow the extracts to diffuse into the media (*Taye et al., 2011*).

We allowed the agar plates to incubate at 37 °C for 24 h. Then we placed gentamycin (30 μg) discs as positive controls on the media and dispensed methanol and petroleum ether (5% DMSO) in the negative control wells. The positive control was selected based on the susceptibility of the bacterium used (*CLSI, 2012*). An agar well (6 mm) showing no zone of inhibition was considered to have no antimicrobial activity. We measured the diameters of the inhibitory growth zones using a ruler and recorded the measurements in millimeters after incubation. We conducted the experiment three times.

## Determination of the minimum inhibitory concentration by the agar dilution method

The minimum inhibitory concentration is the lowest concentration of an antibacterial agent that completely prevents visible growth of the test strain of an organism (*Cock et al., 2022*). For the agar dilution method, plant extracts serially diluted twice in molten MHA and cooled to 50 °C in a water bath to achieve the desired final concentrations. A twofold serial dilution of stock solution containing 200 mg/ml of each plant extract in 5% DMSO used to achieve the desired concentration. The test concentrations were 100 mg/ml, 50 mg/ml, 25 mg/ml, 12.5 mg/ml, 6.25 mg/ml, and 3.12 mg/ml.

The MHA sterilized for 15 min at 121 °C in an autoclave before allowed cooling to 50 °C in a water bath. One milliliter of each twofold serial dilution of plant extract was added to 19 mL of liquefied MHA, which was carefully swirled, poured into sterile petri dishes and allowed to set (*Mazhangara et al., 2020*; *Kowalska-Krochmal & Dudek-Wicher, 2021*).

## Standardization of inocula for agar dilution

A bacterial suspension equivalent to the 0.5 McFarland standard ($1.5 \times 10^8$ CFU/ml) made in sterile saline solution for each ATCC reference. We adjusted the final inoculum for MIC

to $1 \times 10^4$ CFU/spot to get the desired cell density. To obtain the final desired cell density of each bacterial suspension, a standardized inoculum of $10^8$ CFU/ml was further diluted 1:10 in sterile saline solution. Using a micropipette, 0.1 ml of the $10^8$ CFU/ml bacterial inoculum transferred to a sterile Eppendorf tube containing 0.9 ml of sterile saline solution to obtain $10^7$ CFU/ml. By the same method, the inoculum was further diluted to obtain a standard inoculum of $1 \times 10^4$ CFU/spot (*Wiegand, Hilpert & Hancock, 2008*; *Kowalska-Krochmal & Dudek-Wicher, 2021*).

Approximately 1 µL of the prepared inoculum was dispensed on MHA containing different concentrations of plant extracts *via* a 5 µL micropipette. A bacterial suspension was used to inoculate MHAs without plant extract, serving as control plates in the experiment (*Wiegand, Hilpert & Hancock, 2008*). After allowing the spotted inoculum to dry at room temperature for a few minutes, we invert the plates for incubation. The plant extract inhibited visible growth completely at the lowest concentration, which was recorded as the MIC.

## Statistical analysis

All the investigations stored the gathered data in Microsoft Excel 2016 spreadsheets. We conducted all measurements in triplicate to test the reproducibility of the experimental data. We utilized IBM SPSS version 20 software (IBM, Armonk, NY, USA) to perform statistical analysis of the results. A one-way ANOVA, followed by Tukey's *post hoc* multiple comparison test, determined the significance of the difference between the means of the concentrations. The experimental antibacterial activity data express the mean ± standard deviation. A statistical significance level of $P < 0.05$ indicated significance.

## RESULTS

### Percentage yield of the plant extract

The present study presented the percentage yield of methanol extract is 7.85%. We extracted the lowest mass of petroleum ether (6.25%) roots of the plants. Methanol-extracted plants produce a higher yield than petroleum ether-extracted plants.

### *In vitro* antibacterial activity of the plant extract

We evaluated the antibacterial activity of the tuber extract of *E. kebericho* Mesfin through agar well diffusion. We screened methanol and petroleum ether extracts of the plants against four standard bacterial species in triplicate, at concentrations of 200 mg/ml, 100 mg/ml, 50 mg/ml, and 25 mg/ml. The evaluation of antibacterial activity by the agar well dilution method indicated that all the bacteria tested inhibited the growth of the plant extracts of both solvents at a concentration of 200 mg/ml, which ranged from 16.67 ± 0.58 mm to 7.0 ± 0.0 (mm) standard deviation (SD) of the mean inhibition zone diameter, as indicated in Table 1. The observed mean inhibition zone diameters of plant extracts at different concentrations against the tested bacteria were significantly different ($P < 0.05$). The results showed that the extracts have good dose-dependent antibacterial activity.

The maximum activity of the methanol fraction was conferred against *S. aureus* (ATCC 25923) (16.67 ± 0.58 mm), while the minimum activity was observed against *P. aeruginosa*

**Table 1   Mean inhibition zone of *Echinops kebericho* Mesfin tuber against selected bacteria.**

| Solvent | Concentration and controls | Mean inhibition zone of bacterial species (Mean ± Standard deviation (mm)) | | | |
|---|---|---|---|---|---|
| | | *E. coli* | *P. aeruginosa* | *K. pneumonia* | *S. aureus* |
| Methanol | 200 mg/ml | 11.0 ± 1.73[de] | 9.0 ± 1.73[e] | 7.3 ± 0.58[bcde] | 16.67 ± 0.58[bcde] |
| | 100 mg/ml | 9.0 ± 2.64[e] | 7.0 ± 1.0[e] | 6.0 ± 0.0[ae] | 13.3 ± 2.08[acde] |
| | 50 mg/ml | 7.67 ± 2.08[e] | 6.0 ± 0.0[e] | 6.0 ± 0.0[ae] | 8.33 ± 0.58[abe] |
| | 25 mg/ml | 6.0 ± 0.0[ae] | 6.0 ± 0.0[e] | 6.0 ± 0.0[ae] | 6.33 ± 0.58[abe] |
| | Gen 30 µg | 29.67 ± 0.58[abcd] | 29.67 ± 2.08[abcd] | 23.67 ± 0.58[abcd] | 31.33 ± 0.58[abcd] |
| | DMSO 5% | 6.0 ± 0.0[ae] | 6.0 ± 0.0[e] | 6.0 ± 0.0[ae] | 6.0 ± 0.0[abe] |
| P. ether | 200 mg/ml | 8.33 ± 0.58[bcde] | 8.33 ± 0.58[cde] | 7.0 ± 0.0[bcde] | 14.3 ± 1.52[bcde] |
| | 100 mg/ml | 6.5 ± 0.58[ae] | 6.333 ± 0.58[e] | 6.0 ± 0.0[ae] | 10.0 ± 1.0[acde] |
| | 50 mg/ml | 6.0 ± 0.0[ae] | 6.0 ± 0.0[ae] | 6.0 ± 0.0[ae] | 7.3 ± 0.58[abe] |
| | 25 mg/ml | 6.0 ± 0.0[ae] | 6.0 ± 0.0[ae] | 6.0 ± 0.0[ae] | 6.0 ± 0.0[abe] |
| | Gen 30 µg | 29.33 ± 0.58[abcd] | 28.67 ± 0.58[abcd] | 18.33 ± 0.58[abcd] | 30.67 ± 0.58[abcd] |
| | DMSO 5% | 6.0 ± 0.0[ae] | 6.0 ± 0.0[ae] | 6.0 ± 0.0[ae] | 6.0 ± 0.0[abe] |

**Notes.**

Values are expressed as the mean ± SD ($n = 3$). One-way ANOVA followed by Tukey's test was performed; the superscript, [a] compared to 200 mg/ml, [b] to 100 mg/ml, [c] to 50 mg/ml, [d] to 25 mg/ml, [e] to the positive control gentamycin; all superscript letters indicate $P < 0.05$. 6.0 ± 0.0 = no inhibition zone (diameter of agar well). The means of the inhibition zones that do not share superscript letters are not significantly different. (P. Ether) for petroleum ether solvent.

(ATCC 27853) and *K. pneumoniae* (ATCC 43816), with mean inhibition zone diameters of 9.0 ± 1.75 mm and 7.3 ± 0.58 mm, respectively, at the lowest concentration (200 mg/ml), as shown in Table 1. The petroleum ether extract of the plant showed moderate inhibition, with a maximum inhibition of 14.3 ± 1.52 mm against *S. aureus* (ATCC 25923) (Table 1). In general, the extent of bacterial growth inhibition increased as the concentration of the extracts increased. At the concentration of 25 mg/ml, no bacteria inhibited (Table 1). The results obtained for methanol extracts of *E. kebericho* Mesfin showed that there was a significant difference ($P < 0.05$) in the inhibition of all the test organisms at different concentrations, indicating that the antibacterial activities were concentration dependent. The positive control gentamycin strongly inhibited the growth of the selected bacteria.

## Minimum inhibition concentration of the plant extract

The determination of the MIC by agar dilution of the extract revealed *that the extract had an inhibitory effect on S. aureus MIC at a concentration of 25 mg/ml* (ATCC 25923). The MIC of the methanol extract of *E. kebericho* Mesfin against *S. aureus* was the lowest (25 mg/ml) (Table 2). The agar dilution demonstrated that *S. aureus* showed complete growth inhibition at the lowest MIC values compared to the other tested bacteria. The current study confirmed that the methanol extract exhibited the lowest MIC values when compared with the petroleum ether extract.

## Phytochemical constituents of the plant extracts

Phytochemical screening of the methanol and petroleum ether extracts of *E. kebericho* revealed phytochemicals such as alkaloids, flavonoids, tannins, and cardiac glycosides in both solvent extracts (Table 3).

**Table 2  Minimum inhibitory concentrations of plant extracts.**

| Plant extracts | Standard bacterial strains | MIC plant extracts (mg/ml) | |
|---|---|---|---|
| | | MIC of methanol extract | MIC of petroleum ether extract |
| *E. kebericho* | *E. coli* (ATCC 25922) | 50 mg/ml | >100 mg/ml |
| | *S. aureus* (ATCC 25923) | 25 mg/ml | 50 mg/ml |
| | *P. aeruginosa* (ATCC 27853) | 100 mg/ml | 100 mg/ml |
| | *K. pneumoniae* (ATCC 43816) | >100 mg/ml | >100 mg/ml |

**Table 3  Preliminary phytochemical screening of *E. kebericho* with their respective solvents.**

| Test | Plant Extracts | |
|---|---|---|
| | *E. kebericho* (Meth) | *E. kebericho* (P.ether) |
| Alkaloid | + | + |
| Flavonoid | + | + |
| Saponin | − | − |
| Tannin | + | + |
| Phenol | − | − |
| Phytosterol | − | − |
| Cardiac Glycoside | + | + |

**Notes.**
(+) indicates presence and (-) indicates the absence of particular phytochemicals. (Meth) represents methanol, and (P. ether) represents petroleum ether.

## DISCUSSION

Secondary metabolites and phytochemicals produced by plants have a great deal of therapeutic potential as infectious agents (*Sharma et al., 2023*). This study demonstrated the remarkable antibacterial activity of different concentrations of *E. coli*, *P. aeruginosa, K. pneumoniae,* and *S. aureus*. *S. aureus* was sensitive to both methanol and petroleum ether extracts. The variation in the zone of inhibition of methanol and petroleum ether extract varied due to the composition, efficacy, and concentrations of different phytochemicals on the standard reference bacteria (*Alhajali & Ali-Nizam, 2021*).

In the present study, the methanol extract of *E. kebericho* Mesfin inhibited all the studied bacteria at a concentration of 200 mg/ml, with different inhibition ranges. *S. aureus* showed inhibition at the maximum extract concentration, with the greatest inhibition zone. Methanol tuber extract of *E. kebericho* Mesfin exhibited potent inhibitory effects on *S. aureus, K. pneumoniae, P. aeruginosa*, and *E. coli*, consistent with the findings of a previous study conducted by *Ahmed et al. (2023)*. The high antibacterial effect of the methanol extract attributes to the most bioactive compounds. *Deyno et al. (2020)* also reported that *E. kebericho* has antibacterial effects on *S. aureus*. Compared with that of methanol, the petroleum ether extract of *E. kebericho* Mesfin had a minimal inhibition zone, possibly because this bioactive compound is not soluble in nonpolar solvents. In general, gram-negative bacteria showed the least sensitivity to the plant extracts, indicating

bacterial resistance to the plant extracts. The lipophilic or hydrophilic nature of compounds also plays a role in their activity, or lack thereof, against microorganisms. This was to be expected, as gram-negative bacteria have a considerably more complicated barrier system against foreign substance absorption (*Mogana et al., 2020*).

The lowest MIC of the methanol extract ranged from 25 mg/ml (for *S. aureus)* to 100 mg/ml for *P. aeruginosa* and *K. pneumoniae*. The petroleum ether extract of the plant had the lowest MIC, ranging from 50 mg/ml (*S. aureus*) to 100 mg/ml (*P. aeruginosa*), whereas the remaining bacteria did not exhibit activity in the range ≤100 mg/ml. Previous research results agreed with those of the current investigation, in which extraction by different solvents revealed diverse activities and methods used for evaluating MIC differences (*Tariku et al., 2011*). *S. aureus* was more sensitive to both solvent fractions in the overall extract of *E. kebericho* than was the gram-negative organism. The cell wall structure of *S. aureus* makes it more susceptible than gram-negative bacteria. Because gram-negative outer membranes function as diffusion barriers, bacteria are less vulnerable to many antimicrobial treatments (*Sun et al., 2022*).

The current study's findings contradict the MIC values of *K. pneumoniae* from a previous study done by *Deyno et al. (2020)*, which indicated ethyl acetate fraction exhibited *K. pneumonia* (ATCC 43816) and *E. coli* (ATCC 25922). This variation attributed to the solvent used for the extraction and the method employed to evaluate the MIC. Variation among the MICs of plant extracts is due to differences in their chemical composition and volatile constituents of phytochemicals (*Mostafa et al., 2018*).

Preliminary analysis indicated the possible presence of alkaloids, flavonoids, tannins, and cardiac glycosides in both methanol and petroleum ether solvent extracts. Similarly, *Toma et al. (2015)* reported that *E. kebericho* mesfin contains flavonoids, alkaloids, triterpenoids, resins, and steroids. These findings are consistent with previous findings showing that *E. kebericho* tuber extract contains similar bioactive compounds; however, the present findings do not reveal the presence of phenolic compounds, which might be due to variations in the solvent used (*Ameya, Gure & Dessalegn, 2016*; *Yimer et al., 2020*).

## CONCLUSIONS

The results showed that the methanol and petroleum ether extracts of *E. kebericho* Mesfin plants had antibacterial activity against the standard bacterial reference strains *E. coli*, *P. aeruginosa*, *S. aureus*, and *K. pneumoniae*. The methanol extract of the selected plants at a concentration of 200 mg/mL had strong antibacterial effects on *S. aureus*. Additionally, as the concentration of the tested medicinal plant extract increased, the mean diameter of the inhibition zones of the tested bacteria increased proportionally. Compared with the methanol extract, the petroleum ether extract had the least antibacterial activity. Phytochemicals found in the selected medicinal plants included alkaloids, flavonoids, tannins, and other bioactive compounds. The plant extracts in this study have the potential to aid in the development of new drugs and to treat infectious diseases. The antibacterial effects of the selected plants revealed in this study support their use in traditional medicine for treating infectious diseases caused by various bacterial infections in animals and humans.

### Funding
The authors received no funding for this work.

### Competing Interests
The authors declare there are no competing interests.

### Author Contributions

- Jiregna Gari Negasa conceived and designed the experiments, performed the experiments, analyzed the data, prepared figures and/or tables, authored or reviewed drafts of the article, and approved the final draft.
- Ibsa Teshome conceived and designed the experiments, performed the experiments, prepared figures and/or tables, authored or reviewed drafts of the article, and approved the final draft.
- Edilu Jorga Sarba conceived and designed the experiments, analyzed the data, prepared figures and/or tables, authored or reviewed drafts of the article, and approved the final draft.
- Bekiyad Shasho Daro analyzed the data, prepared figures and/or tables, and approved the final draft.

### Data Availability
The raw data are available in the Supplementary Files.

### Supplemental Information
Supplemental information for this article can be found online at http://dx.doi.org/10.7717/peerj.18554#supplemental-information.

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
