# Peer review of "Phytochemical screening and *in vitro* antibacterial activity of *Echinops kebericho* Mesfin tuber extracts: experimental studies"

_PeerJ, doi:10.7717/peerj.18554_

## Round 0.1 · original submission · Major Revisions

Dear Dr. Negasa and colleagues:

Thanks for submitting your manuscript to PeerJ. I have now received two independent reviews of your work, and as you will see, one reviewer raised more concerns about the preliminary nature of the content, particularly the lack of depth with the analyses (e.g., additional data such as chemical profiling would be beneficial in their opinion). Despite this, both reviewers are optimistic about your work and the potential impact it will have on research studying antibacterial activities and potential in medicinal plants. Thus, I encourage you to revise your manuscript, accordingly, considering all the concerns raised by both reviewers.

Fortunately, there is a lot of constructive criticism here for you to consider. The overall presentation of the work (and central thesis) needs to be made clearer. Please ensure that all the figures and tables are necessary (cull redundancy pointed out by the reviewers…use supplemental materials if needed). Your review also seems to be lacking important references from the field. This is surprising for a review on the topic. More recent studies also seem to be missing from the dialogue.

Please note that Reviewer 2 kindly provided a marked-up version of your manuscript.

Thanks again for submitting your work to PeerJ.

Good luck with your revision,

Best,

-joe

Reviewer 1 ·

Basic reporting

The manuscript is fairly well-written but improvements can be made. Some sections are unnecessarily long, such as the introduction and materials and methods sections. Appropriate citations are often sufficient.

Table 1 is redundant. The percentage yield can be cited in the text.
Figure 1 can be included in the supplementary information.
The data for the positive control should be included in Tables 2 and 3.

Experimental design

Standard methodologies were used and sufficient details have been provided. However, there are some issues that need to be addressed.

In the introduction, the authors stated that this species has been traditionally used to treat infections, but are the chosen bacterial species relevant to the traditional medicinal properties?

1. Please explain why methanol and petroleum ether were chosen for extraction, instead of several solvents of different polarity, or to fractionate a single crude extracts into fractions.

2. Please state the name of the botanist who authenticated the plant samples and the location were the specimens were deposited.

3. For the analysis, did the authors performed only one or more independent experiments - apart from triplicates?

4. For the phytochemical testing, the authors should describe clearly how (+) and (++) were assigned based on the outcome of the tests.

Validity of the findings

There is a lack of comparison of the results obtained in this study with the relevant studies that have been reported previously.

There is no clear attempt to correlate the results of the phytochemical tests with those from the antibacterial assays.

Do the authors expect the tubers to contain different array from the aerial parts? This can be discussed.

Conclusions needed to revised to reflect the findings presented in this study.

Line 412-413: ...study provides scientific validation because the bioactive components in plants have highly promising antibacterial effects... - I disagree with this statement as no bioactive compounds have been identified via proper chromatographic techniques, in fact, the nature of the antibacterial compounds in the extracts cannot be deduced based on general phytochemistry tests on crude extracts (without fractionation).

Additional comments

The findings of this study, in general, is rather preliminary due to limited scope of the experimental design. The limitation of this study includes the small number of samples tested (only two solvents for preparation of extracts) and the use of crude extracts instead of fractionated extracts (which allow more insights into the nature of the antibacterial compounds).

Reviewer 2 ·

Basic reporting

Ref: 95466v1
PeerJ
08/04/2024
Title: Phytochemical screening and in vitro antibacterial activity of Echinops kebericho Mesfin tuber extracts: Experimental studies.

Dear Editor,
This manuscript, it is an adequate subject for PeerJ, Where the overall subject is meaningful and worthy of study, However:
Dear authors:
There are some remarks to take into account:

- Abstract : The aim of the research should be better justified, also, the summary is too long, please make it shorter.
- Introduction:
The authors are requested to explain the exact selection of this plant (Echinops kebericho), for study and give some advantages to it. also, the authors should provide some discussion about the advantages of the extraction, phytochemical screening and analysis technique were used.
- In order to give importance to a study on medicinal plants, I suggested the authors add some paragraphs in their manuscript in the introduction about the importance of medicinal plants in human life, and the different uses of medicinal plants.

- The authors are kindly requested to add other biological tests to this extract plant (Echinops kebericho), such as antioxidants, in order to give this study other advantages and to enrich the article.
- The comparison of this data with different previous studies reported in the literature is an important aspect, basing on this, I suggest that the authors add this part their manuscript.
- Is the extraction one or more times in a row and when is the extraction once done as research shows? Can this be considered sufficient for the extraction of active substances in the plant and can it therefore trust the results of the galaxies studied after the extraction process? Because some active substances can remain in the plant - even a few - have a high biological activity?
Therefore, only after improving / incorporating above suggestions the paper should be considered for publication.

Sincerely

Experimental design

Need to revision

Validity of the findings

Need to revision

---

## Round 0.2 · Major Revisions

Dear Dr. Negasa and colleagues:

Thanks for resubmitting your manuscript to PeerJ. I only received one independent review of your revision, and as you will see, this reviewer still suggests substantial revisions. The lack of enthusiasm for your work likely prevented the other two reviewers from helping out, which is unfortunate.

Normally, I would reject your work outright based on these recommendations; however, I am affording you the option of revising your manuscript according to this sole review but understand that your resubmission will likely be sent to at least one new reviewer for a fresh assessment. Of course, if reviewer 1 becomes more positive with your revision, it is possible your work will be much closer to publication.

It appears that, despite the suggestions indicated in previous reviews, you did not support your point of view in a convincing manner. Reviewer 1 believes that your manuscript is still weakly discussed, with many important references are missing. The findings are too preliminary to merit publication currently. The analysis lacks depth. Additional data such as chemical profiling would be very beneficial.

Therefore, I am recommending that you revise your manuscript, accordingly, considering all of the issues raised by the reviewers (from both rounds of review). If your next revision does not address the concerns of the reviewers your work will be rejected.

Good luck with your revision,

-joe

Reviewer 1 ·

Basic reporting

The authors have responded to my queries and agreed with my comments, but did not make the necessary corrections for some comments in the revised manuscript. The rebuttal file seemed incomplete whereby the authors did not provide rebuttals to my other comments under the subsection "Validity of the Findings". Line-by-line rebuttals and citations to changes made in the revised manuscript will make it easier for the reviewers to check if the corrections have been made.

Phytochemical testing is a qualitative assessment but there is still no explanation to differentiate between (+) and (++) in the footnote of Table 4. A mistake was spotted whereby both (+) and (++) denote high presence? Please check.

Table 1 that was suggested to be omitted and the (only) important data (% yield) can be cited in the text. Other data in the table have been stated in the methodology section.

My another concern is the reported values in Table 2 and 3 are relatively high in mg/ml range, and by inference, the antibacterial activities of the extracts of Echinops kebericho tuber are low (or inactive). This has not been sufficiently discussed by the authors. Comparison with relevant literature is also lacking. It is difficult to conclude regarding the claimed antibacterial activities of the tuber extracts.

Language correction is highly recommended.

Experimental design

No comment

Validity of the findings

No comment

Additional comments

Refer to the above.

---

## Round 0.3 · Minor Revisions

Dear Dr. Negasa and colleagues:

Thanks for once again resubmitting your manuscript to PeerJ. Again, reviewer 1 still suggests revisions. Please try to address the suggestions of the reviewer as best as possible.

Good luck with your revision,

-joe

Reviewer 1 ·

Basic reporting

The authors' response to some of the queries are not satisfactory.

1. If the authors decided to use (+) and (-) for results of the phytochemical testing, then the (+) symbol should represent "presence" rather than "high presence" - what does "high presence" means? High compared to? This can be confusing for the readers.

2. The study aims to investigate the antibacterial activity of the extracts but the reported MIC values are in the mg/ml range, mostly above 100 mg/ml. Such high values suggest that the claimed antibacterial activities of the extracts are actually very low/insignificant; some might consider the extracts to be devoid of such bioactivity. This has to be addressed and critically discussed in the manuscript. The authors are suggested to draw comparison to the reported MIC values in the literature. One study reported an MIC value of only around 1 mg/ml (line 390) that is way lower than the values reported in this study. Data should be interpreted objectively.

3. Appropriate comparison is much needed to justify the claims made regarding the potential of the extracts. Words like "high sensitive" (line 360) and "significant antibacterial activity" (line 405) should be used with caution.

4. The discussion should be improved. My suggestion is to draw comparison with relevant literature and to minimize excessive speculation not supported by data.

Experimental design

None

Validity of the findings

None

Additional comments

None

Reviewer 2 ·

Basic reporting

accept

Experimental design

accept

Validity of the findings

accept

Additional comments

accept

---

## Round 0.4 · accepted · Accept

Dear Dr. Negasa and colleagues:

Thanks for revising your manuscript based on the concerns raised by the reviewers. I now believe that your manuscript is suitable for publication. Congratulations! I look forward to seeing this work in print, and I anticipate it being an important resource for groups studying antibacterial activities and potential in medicinal plants. Thanks again for choosing PeerJ to publish such important work.

Best,

-joe